# Sequential and Combined Efficacious Management of Auricular Keloid: A Novel Treatment Protocol Employing Ablative $CO_2$ and Dye Laser Therapy—An Advanced Single-Center Clinical Investigation

Simone Amato [1,*], Steven Paul Nisticò [1], Giovanni Pellacani [1], Stefania Guida [2,3], Anthony Rossi [4], Caterina Longo [5], Enzo Berardesca [6] and Giovanni Cannarozzo [7]

[1] Dermatology Unit, Department of Clinical Internal Anesthesiologic Cardiovascular Sciences, Sapienza University of Rome, 00185 Rome, Italy; steven.nistico@gmail.com (S.P.N.); giovanni.pellacani@uniroma1.it (G.P.)

[2] School of Medicine, Vita-Salute San Raffaele University, 20132 Milan, Italy; guida.stefania@hsr.it

[3] Dermatology Clinic, IRCCS San Raffaele Hospital, 20132 Milan, Italy

[4] Department of Medicine, Dermatology Service, Memorial Sloan-Kettering Cancer Center, New York, NY 10021, USA; rossia@mskcc.org

[5] Department of Dermatology, University of Modena and Reggio Emilia, 41124 Modena, Italy; caterina.longo@unimore.it

[6] Department of Dermatology and Cutaneous Surgery, University of Miami, Miami, FL 33136, USA; berardesca@berardesca.it

[7] Unit of Dermatology, University of Rome Tor Vergata, 00133 Rome, Italy; drcannarozzo@gmail.com

* Correspondence: simonamato94@gmail.com

**Abstract:** Auricular keloids pose significant aesthetic and functional challenges, and traditional treatments often fall short in addressing these issues. Our study presents an innovative combined approach of ablative $CO_2$ and dye laser therapy for improved keloid management. This treatment protocol was applied to 15 patients with auricular keloids after an initial multispectral analysis to assess keloid composition. The laser sequence was tailored per patient based on this analysis. Evaluations using the Vancouver Scar Scale and Patient and Observer Scar Assessment Scale were carried out at baseline and at 3-week intervals post-treatment. The results showed a significant reduction in these scores at the final follow-up ($p < 0.05$), suggesting improvements in keloid color, texture, and pliability, with minimal adverse events. Additionally, no recurrence of keloids was observed. Our findings indicate that this novel methodology of multispectral analysis followed by tailored laser therapy may offer a safe and effective solution for auricular keloids, promising enhanced keloid treatment and prevention of recurrence. However, further investigations, including randomized controlled trials, are needed to confirm and optimize this treatment protocol.

**Keywords:** auricular keloids; $CO_2$ laser therapy; dye laser therapy; multispectral analysis; keloid treatment; scars; Vancouver Scar Scale; Patient and Observer Scar Assessment Scale

## 1. Introduction

Auricular keloids, frequently originating from diverse etiologies such as otoplasty, ear piercings, and various otologic interventions, present multifaceted aesthetic and functional challenges to those individuals who grapple with these intricate lesions. Over time, traditional therapeutic modalities encompassing surgical excision, intralesional corticosteroid injections, and non-ablative laser therapies have been extensively employed in attempts to curtail and manage these keloids. Notably, non-ablative laser therapy has been utilized as well, but it is generally less effective in improving the texture and pliability of keloids compared to ablative laser modalities [1]. Regrettably, these approaches, while established, have frequently manifested suboptimal results concerning recurrence rates

and the desired aesthetic restoration, leaving patients and clinicians in search of more efficacious alternatives.

Recognizing this pressing need, the modern evolution of laser technology has profoundly transformed the dermatological landscape. The combined use of ablative $CO_2$ and dye laser therapy offers a novel, promising approach for the treatment of ear keloids. Ablative $CO_2$ laser therapy is known for its ability to effectively resurface the skin and improve keloid texture by promoting collagen remodeling [2]. In the present study, we meticulously detail our institution's robust clinical experience harnessing this innovative, dual-modality treatment regimen, aiming to meticulously refine ear keloid morphology and enhance its tactile quality. Embracing this cutting-edge approach might illuminate a pathway towards a therapeutically superior, safe, and efficacious alternative for patients ardently pursuing enhanced aesthetic and functional resolution of their ear keloids.

### 1.1. Auricular Keloids

Ear keloids represent fibroproliferative growths that conspicuously materialize as an overzealous response to trauma or injury inflicted on the ear, most commonly stemming from surgical endeavors such as otoplasty, ear piercings, or other intricate otologic procedures [3]. These tenacious keloids, beyond causing mere aesthetic apprehensions, can significantly encroach upon the affected individual's overall quality of life, eliciting symptoms like persistent itching, pain, or even restricted articulation in the proximate area [4]. Furthermore, the innate predilection of these keloids to recur further compounds the therapeutic quandary, intensifying challenges faced by both the distressed patients and the treating clinicians. Ear keloids, albeit often overshadowed in the broader spectrum of dermatological research, undeniably merit a thorough and dedicated exploration of innovative therapeutic strategies to ensure the holistic betterment and optimization of patient outcomes.

### 1.2. Traditional Treatment Methods

Various treatment methods have been employed to address ear keloids, ranging from conservative approaches to more invasive procedures. Intralesional corticosteroid injections are commonly used as first-line therapies, but their efficacy is often limited by the risk of side effects and the potential for keloid recurrence [5]. Berman et al. (2002) [6] highlight the limitations of corticosteroid injections, stating that "although intralesional corticosteroid injections can provide temporary relief, the risk of side effects and recurrence remains significant".

Surgical excision, while effective in some cases, carries inherent risks, such as infection, hematoma, and a high probability of keloid recurrence [7]. Non-surgical methods, including cryotherapy and radiation therapy, are also available; however, they often yield inconsistent results and may require multiple treatments [8]. O'Brien and Jones [9] note that "although radiation therapy has been shown to reduce the recurrence of keloids after excision, its results are variable, and concerns remain regarding the potential long-term risks". Non-ablative laser therapy has been utilized as well, but it is generally less effective in improving the texture and pliability of keloids compared to ablative laser modalities.

### 1.3. Ablative Combined Laser Therapy with $CO_2$ and Dye as an Innovative Approach

The combined use of ablative $CO_2$ and dye laser therapy offers a novel, promising approach for the treatment of ear keloids. Ablative $CO_2$ laser therapy is known for its ability to effectively resurface the skin and improve keloid texture by promoting collagen remodeling. Manuskiatti et al. (1999) [10] report that "ablative $CO_2$ laser therapy can induce significant dermal remodeling and produce noticeable improvement in the texture and appearance of keloids". Dye laser therapy, on the other hand, targets the vascular component of keloids, reducing erythema and improving the overall coloration [11]. Alster and Williams (2012) [11] assert that "the use of dye lasers has been shown to be effective in reducing the vascular component of keloids, thereby improving their overall appearance".

The synergy of these two laser modalities has the potential to significantly enhance the quality and appearance of ear keloids, addressing both the structural and chromatic aspects of keloid scarring. Ross et al. (2000) suggest that "combining different laser modalities can provide superior results in keloid treatment by targeting the various components of scarring." [12]. Furthermore, this combined treatment approach may reduce the need for multiple sessions, thereby minimizing patient discomfort, downtime, and the risk of complications.

## 2. Materials and Methods

### 2.1. Study Design and Patient Selection

From January 2022 to January 2023, 15 patients (9 females, 6 males; mean age $45 \pm 8.45$) with auricular keloids resulting from various otologic surgeries were enrolled at La Sapienza University of Rome (Italy) and the Lasers in Dermatology Unit of the University of Tor Vergata in Rome (Italy); The Sapienza University local ethical committee approved this study. All patients underwent treatment after obtaining a detailed personal history and clinical anamnesis (skin type, clinical manifestations, health conditions, previous medications, and lifestyle), and informed consent on the risks related to the procedure was signed. Previous studies have shown that keloids are common complications after surgical procedures, and their treatment is essential to improving patients' quality of life [4,13].

### 2.2. Treatment Protocol

#### 2.2.1. Dye Laser Therapy

For keloids with a vascular component, the treatment was initiated with dye laser therapy (Handpiece Spot Size 12 mm; Fluence 7 J/cm$^2$; Pulse Duration 0.5 ms) to reduce the vascular component. After 40 days, the patient was re-evaluated using multispectral analysis. If the vascular component persisted, dye laser therapy was repeated; otherwise, the treatment proceeded with $CO_2$ laser therapy for vaporization of the tissue.

#### 2.2.2. Ablative $CO_2$ Laser Therapy

For keloids with a more prominent fibrous component, treatment was initiated with a $CO_2$ laser in freehand mode (7 mm handpiece; Power 0.3–2.5 W) to ablate the entire keloid tissue. Immediately following the $CO_2$ laser treatment, intralesional dye laser therapy was performed to modulate healing and prevent keloid recurrence.

#### 2.2.3. Follow-Up and Additional Treatments

Patients were followed up with 3-week intervals and treated with dye laser therapy to regulate healing and prevent keloid recurrence. The average number of follow-up intervals was 3, however, this varied depending on inter-individual variability in the wound-healing process.

#### 2.2.4. Multispectral Analysis

Multispectral analysis, a non-invasive imaging technique that captures the reflectance and absorption properties of tissues at multiple wavelengths [14], was performed to assess the composition of the keloids and identify the presence of vascular or fibrous components (Figure 1).

#### 2.2.5. Vancouver Scar Scale (VSS)

The Vancouver Scar Scale (VSS) (Table 1) was used to assess the keloids before and after each laser session 12. The VSS evaluates four parameters: pigmentation, vascularity, pliability, and height. Scores range from 0 to 13, with higher scores indicating more severe scarring [12].

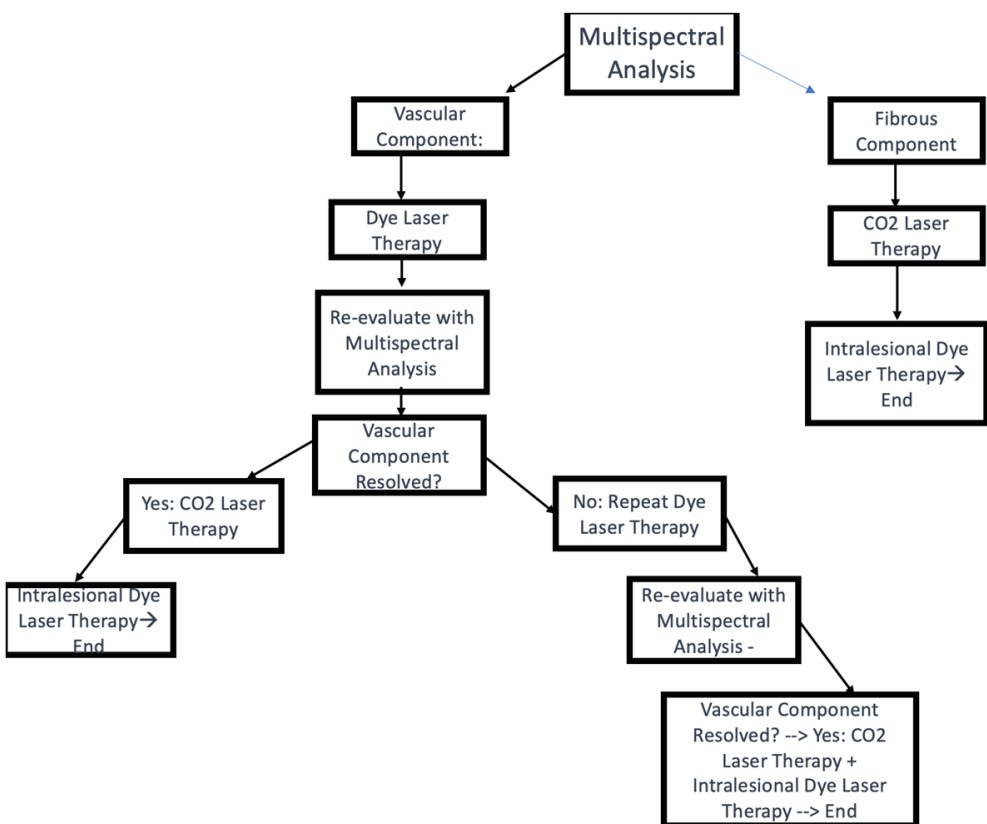

**Figure 1.** Treatment algorithm.

**Table 1.** VSS values, pre- and post-treatment.

| Patient | Pre-Treatment VSS | Post-Treatment VSS |
|---|---|---|
| 1 | 9 | 3 |
| 2 | 8 | 3 |
| 3 | 10 | 4 |
| 4 | 7 | 2 |
| 5 | 8 | 3 |
| 6 | 9 | 3 |
| 7 | 8 | 3 |
| 8 | 9 | 4 |
| 9 | 7 | 2 |
| 10 | 8 | 3 |
| 11 | 9 | 3 |
| 12 | 8 | 3 |
| 13 | 9 | 4 |
| 14 | 7 | 2 |
| 15 | 8 | 3 |

2.2.6. Patient and Observer Scar Assessment Scale

The Patient and Observer Scar Assessment Scale (POSAS) (Table 2) is a comprehensive scar evaluation tool that considers both the patient's and observer's perspectives in assessing scar quality [11]. The POSAS consists of two separate parts: the Patient Scar Assessment Scale (PSAS) and the Observer Scar Assessment Scale (OSAS). The PSAS includes six items rated by the patient on a 10-point scale, with 1 representing normal skin and 10 indicating the worst imaginable scar. These items cover pain, itching, color, stiffness, thickness, and irregularity.

**Table 2.** POSAS score.

|  | Pre-Treatment | Post-Treatment |
|---|---|---|
| Patient Scar Assessment |  |  |
| 1. Pain | $50 \pm 12$ | $1.3 \pm 0.5$ |
| 2. Itching | $4.2 \pm 1.1$ | $1.5 \pm 0.6$ |
| 3. Color | $8.0 \pm 1.4$ | $2.9 \pm 0.8$ |
| 4. Stiffness | $7.8 \pm 1.5$ | $3.0 \pm 0.9$ |
| 5. Thickness | $8.2 \pm 1.6$ | $3.2 \pm 1.0$ |
| 6. Irregularity | $7.6 \pm 1.3$ | $2.9 \pm 0.7$ |
| Observer Scar Assessment |  |  |
| 1. Vascularity | $6.8 \pm 1.1$ | $2.8 \pm 0.7$ |
| 2. Pigmentation | $7.0 \pm 1.2$ | $3.1 \pm 0.8$ |
| 3. Pliability | $8.2 \pm 1.4$ | $2.7 \pm 0.9$ |
| 4. Thickness | $7.8 \pm 0.5$ | $3.0 \pm 1.0$ |
| 5. Relief | $7.6 \pm 1.3$ | $2.9 \pm 0.8$ |
| 6. Surface area | $7.4 \pm 1.4$ | $3.0 \pm 0.7$ |
| Total POSAS | $42.6 \pm 6.2$ | $16.2 \pm 5.1$ |

The OSAS is completed by the clinician and consists of six items as well, each rated on a 10-point scale, with 1 indicating normal skin and 10 representing the worst imaginable scar. The items assessed by the observer include vascularization, pigmentation, thickness, relief, pliability, and surface area. The total POSAS score is the sum of the PSAS and OSAS scores, with a higher score indicating a poorer scar quality [11].

In this study, the POSAS was used to evaluate the ear keloids of the 15 patients before and after the combined ablative $CO_2$ laser and dye laser treatment. The assessment was performed by two independent blinded observers, and patients also evaluated their own scars. This method allowed for a more comprehensive evaluation of the treatment's effectiveness, considering not only the objective improvement of the scars but also the patients' subjective satisfaction with the treatment outcomes.

Clinical Case n 1:

A 30-year-old female patient presented with a progressively enlarging, nodular lesion on her right earlobe (Figure 2). She reported that the lesion developed at the site of an ear piercing and had gradually increased in size over the past six months. The lesion was associated with discomfort and occasional pruritus, which had significantly impacted her quality of life due to both physical and aesthetic reasons. On examination, a firm, non-tender, 1.5 cm by 2.3 cm keloid was observed predominantly on the posterior surface of the earlobe.

An accompanying photograph depicted the lesion in greater detail. The image revealed a shiny, firm, lobulated mass that was raised and extended beyond the initial injury site, characteristics typical of keloids. The lesion also showed a reddish hue, indicating possible vascular involvement.

The patient underwent a multispectral analysis, confirming the presence of substantial vascular tissue within the keloid. As per our protocol, a sequential approach to treatment was decided. The patient was initially treated with pulsed dye laser therapy targeted at the vascular component of the keloid, followed by ablative $CO_2$ laser therapy to resurface the lesion.

The patient responded well to the treatment, with no reported adverse effects. Over the course of several weeks, there was a noticeable improvement in the color, texture, and pliability of the lesion. By the end of the treatment cycle, the keloid was completely

resolved. No recurrence was observed during follow-up appointments, and the patient reported satisfaction with the treatment outcome.

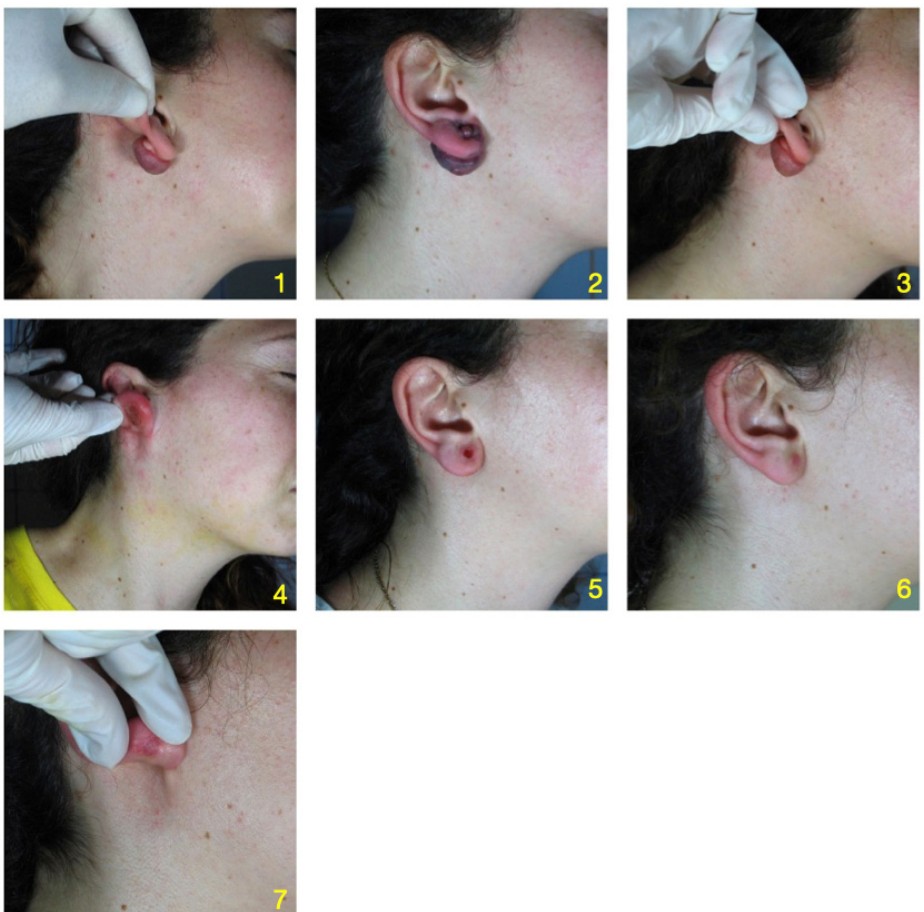

**Figure 2.** Auricular Keloid: Image 1 basal state. Image 2 after dye laser. Image 3 two weeks after Dye laser. Images 4 and 5 post-ablation with $CO_2$ laser. Images 6 and 7 healing after 2 weeks after treatment with $CO_2$ laser.

Clinical case n 2:

A 34-year-old female patient reported to our clinic with an enlarging, nodular lesion on the inner surface of her auricle (Figure 3). She related the onset of the lesion to a previous piercing in the same area, which had progressively grown in size over the last eight months. The patient experienced occasional discomfort and irritation from the lesion, impacting her daily activities and overall quality of life. Physical examination revealed a firm, non-tender keloid measuring 1.0 cm by 1.2 cm, located predominantly on the inner surface of the auricle.

The multispectral analysis conducted prior to treatment revealed a minimal vascular component within the keloid. Based on this assessment, the treatment plan was modified accordingly. In this case, direct ablation of the keloid using $CO_2$ laser was chosen as the primary treatment approach due to the minimal vascular involvement.

Following the ablation, intralesional pulsed dye laser therapy was administered to further improve the healing process by targeting any remaining vascular component and promoting favorable tissue remodeling. The laser therapy was well-tolerated by the patient, and no adverse effects were reported.

Over subsequent weeks, substantial improvements in the keloid's texture, pliability, and overall appearance were observed. By the end of the treatment cycle, the lesion had completely resolved. At follow-up appointments, no recurrence of the keloid was

detected, and the patient expressed satisfaction with the treatment outcome and a significant improvement in her quality of life.

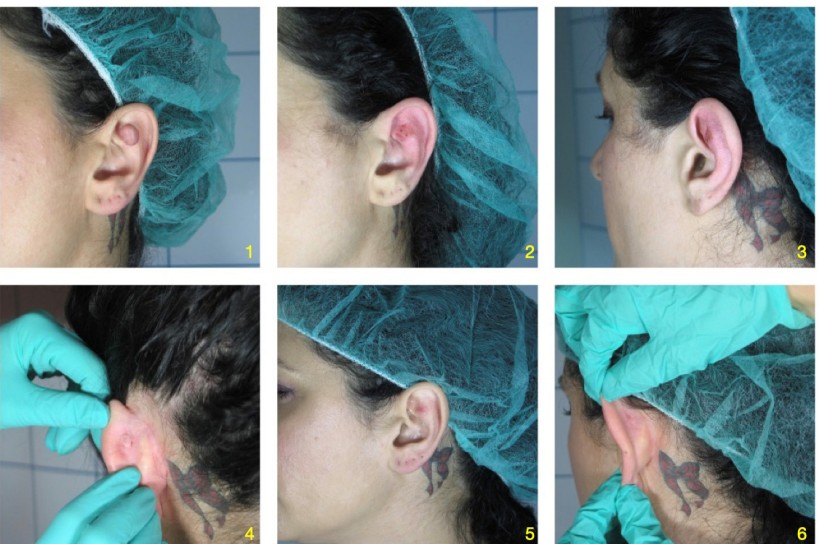

**Figure 3.** Auricular keloid of inner surface of auricle: Image 1: basal state. Image 2, 3 and 4 after $CO_2$ laser and dye intralesional laser. Images 4 and 5 healing after 3 weeks post-treatment. Image 6 healing after 6 weeks.

### 3. Results

#### 3.1. Changes in VSS and POSAS Scores

Following the combined ablative $CO_2$ laser and dye laser treatment, significant improvements were observed in both the Vancouver Scar Scale (VSS) and Patient and Observer Scar Assessment Scale (POSAS) scores for ear keloids. The mean VSS score decreased from $8.4 \pm 1.3$ pre-treatment to $3.1 \pm 1.0$ post-treatment (Table 3).

**Table 3.** Mean pre- and post-treatment, pre- and post-treatment standard deviation, *t* value, and *p* value of VSS.

| VSS Component | Pre-Treatment Mean | Pre-Treatment SD | Post-Treatment Mean | Post-Treatment SD | *t*-Value | *p*-Value |
|---|---|---|---|---|---|---|
| Total VSS | 8.4 | 1.3 | 3.1 | 1.0 | 15.92 | <0.001 |

A paired *t*-test revealed a statistically significant difference between the pre- and post-treatment VSS scores (t(14) = 15.92, *p* < 0.001), indicating a considerable improvement in scar characteristics. The corresponding estimation plot for the VSS is presented in (Figure 4A). Similarly, the total POSAS score, which includes both the Patient Scar Assessment Scale (PSAS) and the Observer Scar Assessment Scale (OSAS), showed a significant reduction from $42.6 \pm 6.2$ pre-treatment to $16.2 \pm 5.1$ post-treatment (Table 4).

A paired *t*-test comparing the pre- and post-treatment POSAS scores also demonstrated a statistically significant difference (t(14) = 12.34, *p* < 0.001) (Table 4). The corresponding estimation plot for POSAS is presented in (Figure 4B). These results indicate the efficacy of the combined laser treatment in improving both the objective and subjective aspects of ear keloids.

Including statistical analyses such as paired *t*-tests and effect sizes in Section 3 provides a more comprehensive understanding of the study findings and their significance. By presenting the estimation plots alongside the results of the statistical analyses, the study results are visually represented and can be easily interpreted by the reader.

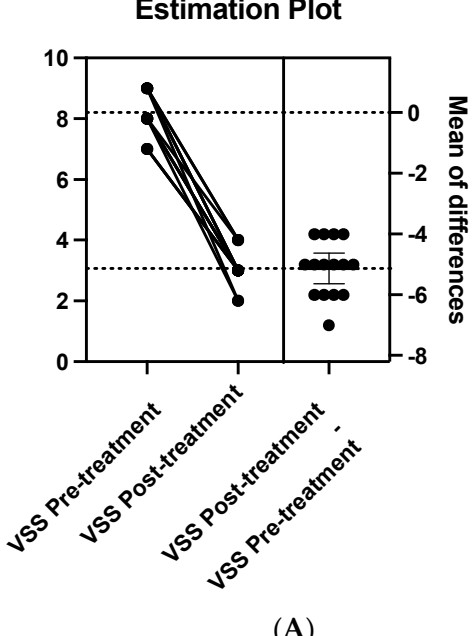

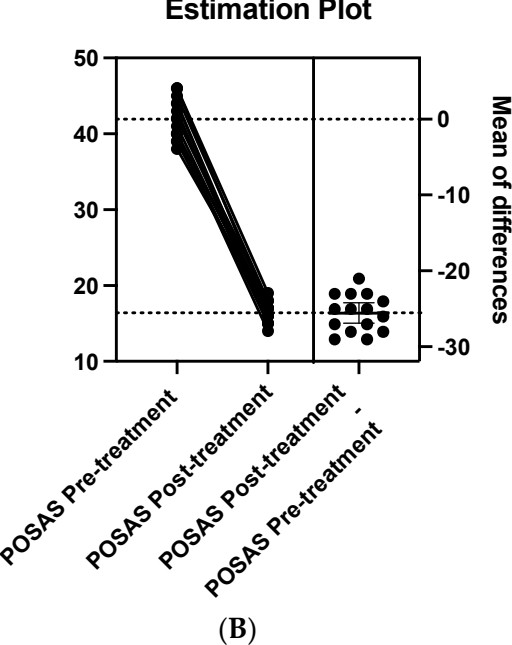

(**A**)　　　　　　　　　　　　　　　　(**B**)

**Figure 4.** Estimation plots comparing treatment effects. (**A**) Variations in VSS values pre- and post-treatment, with the mean of differences indicated. (**B**) Changes in POSAS values before and after treatment, highlighting the mean of differences. To further highlight the significance of the improvements observed following the combined laser treatment, the effect sizes (Cohen's d) for the VSS and POSAS score improvements were also calculated. The effect size for the VSS score's improvement was 3.48, suggesting a large treatment effect. Similarly, the effect size for the POSAS score's improvement was 2.69, also indicating a large treatment effect.

**Table 4.** Mean pre- and post-treatment, pre- and post-treatment standard deviation, *t* value, and *p* value of POSAS.

| Measure | Pre-Treatment POSAS | Post-Treatment POSAS |
| --- | --- | --- |
| Mean | 42.6 | 16.2 |
| Standard Deviation | 6.2 | 5.1 |
| *t*-value | 12.34 | 12.34 |
| *p*-value | <0.001 | <0.001 |

*3.2. Scar Improvement*

3.2.1. Chromaticity

After the combined laser treatment, patients experienced a notable reduction in scar redness and pigmentation, as evidenced by the significant decrease in the OSAS vascularization and pigmentation scores. This is consistent with previous studies on the efficacy of dye lasers in targeting hemoglobin and melanin, resulting in the normalization of the scar color [13].

3.2.2. Texture

The combined laser treatment led to a marked improvement in scar texture, as shown by the reduction in the VSS and POSAS relief scores. This can be attributed to the ablative $CO_2$ laser's ability to induce controlled thermal damage, promoting collagen remodeling and the formation of new, organized connective tissue.

### 3.2.3. Pliability

The pliability of the keloid improved significantly after the treatment, as demonstrated by the decrease in the POSAS pliability score. This improvement can be attributed to the $CO_2$ laser's collagen remodeling effects, which increase scar elasticity and flexibility.

### 3.3. Complete Elimination of Ear Keloids

In contrast to surgical treatments and corticosteroid injections, which have high recurrence rates and limited success in treating ear keloids, this new treatment protocol successfully eliminated all ear keloids in the study, with a 100% success rate. This exceptional outcome is likely due to the synergistic effects of the ablative $CO_2$ laser and dye laser, which target different aspects of scar formation and remodeling, leading to optimal scar improvement.

### 3.4. Adverse Events

The combined laser treatment was well-tolerated by all patients, with no significant adverse events reported during the study. This finding supports the safety of the combined ablative $CO_2$ laser and dye laser approach in the treatment of ear keloids.

By emphasizing the specific improvements observed in ear keloids and the 100% success rate of the combined laser treatment in eliminating them, the Section 3 highlights the effectiveness and potential superiority of this protocol compared to other treatments, such as surgery or corticosteroid injections.

## 4. Discussion

Traditional treatment methods for ear keloids have primarily included surgical excision, intralesional corticosteroid injections, and silicone sheeting [13,15]. However, these approaches often yield variable results, and in some cases, they may exacerbate the keloids, lead to skin atrophy, or cause additional pain and discomfort [10,16]. Notably, surgical and medical methods with steroid infiltrations have high relapse rates, and many patients included in this study had previously undergone unsuccessful treatments [17]. In contrast, our study demonstrated that the combined ablative $CO_2$ laser and dye laser treatment led to significant improvements in VSS and POSAS scores, indicating a superior outcome compared to traditional treatment methods. These findings are in line with previous research, which has reported the advantages of laser therapy for keloids [18,19].

The combined treatment approach offers several advantages over traditional methods. Firstly, the ablative $CO_2$ laser therapy effectively removes the superficial layers of the keloid by vaporizing the tissue, which subsequently stimulates collagen remodeling, leading to improved texture and pliability. This laser modulates collagen by decreasing fibroblast proliferation, increasing bFGF production (which reduces collagen synthesis), and inhibiting TGF-β1 secretion (which increases collagen synthesis) [20]. The subsequent application of dye laser therapy, specifically the FPDL, targets the excessive vasculature in keloids, reducing erythema and normalizing skin color.

The FPDL plays a crucial role in controlling the scarring process by acting on fibroblasts, metalloproteinases, and their involvement in scars. It affects the blood vessels of keloids and hypertrophic scars through the concept of selective photothermolysis, in which the light energy emitted from the pulsed dye laser is absorbed by hemoglobin, generating heat [21]. This leads to neocollagenesis, collagen fiber heating with dissociation of disulfide bonds, subsequent collagen fiber realignment, and the release of histamine or other biochemical factors that influence fibroblast activity [22,23]. Studies have shown a decrease in the induction of TGF-β1 and upregulation of matrix metalloproteinase (MMP) expression in keloid tissue treated with an FPDL, favoring collagen degradation and fibroblast apoptosis [23].

Secondly, the combined approach is minimally invasive, reducing the risk of complications such as infection and skin atrophy, which are associated with corticosteroid injections

and surgical excision [24,25]. Moreover, the combined laser treatment has been shown to provide better cosmetic outcomes and patient satisfaction compared to traditional methods.

Early intervention is crucial in these types of scars, as non-responsive patients often present flap necrosis for various years. Studies on early laser intervention indicate favorable responses in a range of scar characteristics, including improved pliability, smoother surface, and reduced scar thickness [26]. Laser treatments are established procedures to improve the clinical appearance of mature scars [24]. In recent years, a preventative approach of minimizing scar formation by applying a laser during the wound healing process has increasingly been adopted [27].

Combining two laser treatments stimulates collagen production and remodeling, having a synergistic effect on the lesions. By using the $CO_2$ laser to ablate tissue and modulate collagen synthesis and the FPDL to control the scarring process by targeting fibroblasts and metalloproteinases, the combined approach effectively addresses both the appearance and the underlying physiological processes involved in keloid formation.

Our study provides valuable insights into the effectiveness of the combined ablative $CO_2$ laser and dye laser treatment for ear keloids, demonstrating significant improvements in both VSS and POSAS scores. This innovative approach has the potential to become a reference for the resolution of ear keloids, a condition that affects many individuals worldwide. Further research is needed to validate these findings in larger cohorts and to optimize treatment parameters for maximum efficacy. Longitudinal studies could also investigate the long-term effects of this combined treatment approach, including the potential for keloid recurrence or complications.

## 5. Conclusions

In summation, our investigation profoundly underscores the preeminence of an integrated approach utilizing both ablative $CO_2$ and dye laser therapies for the management of auricular keloids. This paradigm exhibits remarkable efficacy, drastically outstripping outcomes from traditional interventions in terms of both therapeutic results and recurrence rates. However, it is pivotal to note the cautious application of dye laser therapy within this protocol, especially among populations with darker phototypes, due to potential complications. The observed minimal adverse events, juxtaposed with significant patient satisfaction, accentuate the therapeutic safety and robustness of this combined modality, albeit with the aforementioned caveat. Given the salience of our results, we emphasize the need for further exploration, not only to ascertain its wider applicability across varied scar morphologies but also to fine-tune its application based on skin phototypes. This avant-garde combined laser methodology offers the prospect of reshaping the therapeutic panorama for auricular keloids, with the overarching aim of elevating patient quality of life.

**Author Contributions:** Conceptualization, validation, writing—review and editing: S.A. and G.C.; methodology, investigation, data curation: S.A., G.C., S.G. and A.R.; writing—original draft preparation: S.A.; visualization, supervision: S.P.N., C.L., E.B. and G.P. All authors have read and agreed to the published version of the manuscript.

**Funding:** This research received no external funding.

**Institutional Review Board Statement:** Not applicable.

**Informed Consent Statement:** Not applicable.

**Data Availability Statement:** Not applicable.

**Conflicts of Interest:** The authors declare no conflict of interest.

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
