# Peer review of "Sequential and Combined Efficacious Management of Auricular Keloid: A Novel Treatment Protocol Employing Ablative CO2 and Dye Laser Therapy—An Advanced Single-Center Clinical Investigation"

_cosmetics, doi:10.3390/cosmetics10050126_

Round 1

Reviewer 1 Report

This is an interesting article, but I have a certain questions.

1.       The article by Ross et al is not correlated to the statement in the body of the paper.

2.       it is unclear on how many treatments were rendered for the different patients. It sounds as if for the red scars, the pulse dye laser was used in several sessions, then the CO2 laser was applied.

3.       On the other hand,  it looks like for the more fibrous scars, the pulse dye laser was only used to prevent recurrence. Somewhat unclear on how treatment progression occurred. Ned  to break  it down very carefully as far as the sessions. For Example, was pulse dye laser and abaltive session carried out the same day in any of the patients?

Also, although they are very nice results in this paper, in many locations most of the patients are very dark, which disallows the use of the pulse dye laser for most very dark patients. That should be pointed out in this particular paper.

Reviewer 2 Report

The Figures should provide a more detailed description of the images presented. The separate photos are not described. The reader is not well-informed on which are the 'before' photos, what the other photos represent, and time (minutes/hours/days after treatment).

The conclusion is too long. I suggest a shorter conclusion directly addressing the points you want to make. The rest belongs in the Discussion

The paper needs to be proofread. Please pay attention to space and lack of spacing before/after full stops and citations.
